# Microbiota Profiling on Veterinary Faculty Restroom Surfaces and Source Tracking

**DOI:** 10.3390/microorganisms11082053

**Published:** 2023-08-10

**Authors:** Hiba Jabri, Simone Krings, Papa Abdoulaye Fall, Denis Baurain, Georges Daube, Bernard Taminiau

**Affiliations:** 1Laboratory of Food Microbiology, Fundamental and Applied Research for Animals and Health Center (FARAH), Department of Food Sciences, Faculty of Veterinary Medicine, University of Liege, Quartier Vallée 2, B42, Avenue de Cureghem 10, 4000 Liège, Belgium; hibajabri88@hotmail.fr (H.J.); georges.daube@uliege.be (G.D.); 2Department of Microbial Sciences, School of Biosciences, Faculty of Health and Medical Sciences, Univesity of Surrey, Guildford GU2 7XH, UK; sk01450@surrey.ac.uk; 3FoodChainID GENOMICS, Laboratory Manager NGS, Rue Hayeneux, 62, 4040 Herstal, Belgium; abdoulaye.fall@foodchainid.com; 4Eukaryotic Phylogenomics, InBioS-PhytoSYSTEMS, University of Liège, 4000 Liège, Belgium; denis.baurain@uliege.be

**Keywords:** microorganisms, biotopes, 16S rDNA amplicon sequencing, database, restrooms, microbial source tracking

## Abstract

In this study, we aimed to develop a comprehensive microbial source amplicon database tailored for source tracking in veterinary settings. We rigorously tested our locally curated source tracking database by selecting a frequently accessed environment by veterinary students and veterinarians. By exploring the composition of resident microbiota and identifying potential sources of contamination, including animals, the environment, and human beings, we aimed to provide valuable insights into the dynamics of microbial transmission within veterinary facilities. The 16S rDNA amplicon sequencing was used to determine the bacterial taxonomic profiles of restroom surfaces. Bacterial sources were identified by linking our metadata-enriched local database to the microbiota profiling analysis using high-quality sequences. Microbiota profiling shows the dominance of four phyla: Actinobacteria, Bacteroidetes, Proteobacteria, and Firmicutes. If the restroom cleaning process did not appear to impact microbiota composition, significant differences regarding bacterial distribution were observed between male and female users in different sampling campaigns. Combining 16S rDNA profiling to our specific sources labeling pipeline, we found aquatic and human sources were the primary environment keywords in our campaigns. The probable presence of known animal sources (bovids, insects, equids, suids…) associated with bacterial genera such as *Chryseobacterium*, *Bergeyella*, *Fibrobacter*, and *Syntrophococcus* was also involved in restroom surfaces, emphasizing the proximity between these restrooms and the exchange of bacteria between people involved in animals handling. To summarize, we have demonstrated that DNA sequence-based source tracking may be integrated with high-throughput bacterial community analysis to enrich microbial investigation of potential bacterial contamination sources, especially for little known or poorly identified taxa. However, more research is needed to determine the tool’s utility in other applications.

## 1. Introduction

In the veterinary environment, most animals can be considered potential carriers and probably risk transmitters of bacterial pathogens by professionals. For example, zoonotic bacteria can be transferred from animals to human beings in several ways, including direct and indirect contact with a veterinarian. Buildings with high human activity in the veterinary environments (students or veterinarians) such as restrooms and nests for bacterial exchange and dissemination (human skin, water, soil, or also animal source). Students or veterinarians can be vectors for various pathogenic bacteria, not only for themselves but also for the animals they come in contact with. Restrooms have always been regarded as potential sources of infectious diseases. Bacterial pathogens such as some strains of *E. coli*, which are often found in restrooms, can be transferred by hands, gowns, and boots to cattle, pigs, horses, dogs, or cats in a veterinary environment [1]. *Staphylococcus aureus (S. aureus)* strains were isolated from cows with mastitis, from horses and dogs with lesions, from human beings, from dogs and cats who were healthy carriers [2], and could also be found in restrooms. Transmission of *S. aureus* between human beings and animals has also been reported [3,4].

It has been demonstrated that human feces can carry a wide range of transmissible pathogens: *Campylobacter*, enterohaemorrhagic *Escherichia coli*, *Salmonella*, *Shigella*, *Staphylococcus*, and *Yersinia* as well as viruses such as norovirus, rotavirus, and hepatitis A and E, just to name a few [5]. Fecal indicator bacteria (FIB) could be used as a marker of fecal pollution and an indication of the pathogen population [6]. However, in terms of source tracking, FIB are members of bacterial groups or taxa that are ubiquitous in human and other animal feces. Therefore, they provide little or no information about specific contaminating hosts. Little is known about the other possible contaminations, either direct or indirect, from environmental or animal sources. Bacterial biogeography is mainly performed with a bacterial identification campaign in a given biotope. With this strategy, the link between microbes and biotopes can only be achieved with proper taxonomical identification, leaving out unknown bacterial populations. Moreover, uncultivable bacteria can also be ruled out if this identification process relies only on microbial culture. Additionally, linking bacteria with biotopes can be further characterized by directly linking bacterial genetic fingerprints with biogeography, even for uncharacterized or unknown populations.

Comparison of collected sequence data (CCSD) approach using a given phylogenetic target (e.g., the 16S rDNA) with existing datasets in genetic databases (using the same sequencing technologies) has already been tested and could be used as a start to identify bacterial environmental origin for anthropogenic microbial communities (e.g., human skin, soil, etc.) [7]. So far, published CCSD campaigns have been restricted to include only well-described bacterial populations and focused mainly on human-associated biotopes. The CCSD approach can be used to detect the probable bacterial contaminants in restroom environments, especially those of animal origin by adding a data connection between bacteria and eukaryotic sources (animals or plants). There is surprisingly no comprehensive database for bacterial biogeography nor a database linking eukaryote organisms as hosts for bacteria. If such databases exist, could they be a support to help us to improve source tracking studies for little known or poorly identified microbial contamination taxa?

To address this question, we designed a sampling campaign targeting restrooms considering criteria like gender, surfaces, and hygiene influence. We created a database where 16S rDNA sequences referenced in the public rDNA database SILVA v.132 [8] were linked to source metadata encoded using a controlled vocabulary (CV). This CV was constructed with an environmental annotation ontology model by adding for the first time the eukaryotic taxonomic classification for their probable host sources organisms using the vernacular term. This allowed us to enrich the microbiota profiling campaign of restrooms with probable sources of contamination in restrooms.

## 2. Materials and Methods

### 2.1. Sampled Surfaces

All samples were collected in three sampling campaigns on different dates (March 2017, March 2018, and April 2018) from two restrooms used by potentially 100 to 150 veterinarian students per day, one used by women (*n* = 48 samples) and one by men (*n* = 48 samples). Eight surfaces (door handles inside and outside of the restroom, handles inside and outside of the toilet cabin, tap faucet handle, toilet seat, toilet flush handle, and urinal flush used by men’s restroom) were sampled in the restroom evenly distributed in the same buildings at the Faculty of Veterinary Medicine in University of Liège, Belgium. Samples for each surface were taken considering two criteria of hygiene: one after the cleaning hygiene process (Clean) and the other before (Dirty) (Table 1). We replicated the samples three times.

In the cleaning process, a Sani Cud Pur Eco product from the brand Diversey was utilized, containing citric acid and surfactant agents. The cleaner applied the product and added water to effectively clean the restroom. It is essential to clarify that our emphasis on the cleaning hygiene process was to enable a diverse range of sample testing, rather than specifically evaluating its effect on microbial composition. Our primary objective was centered on exploring potential shifts in microbial sources, with a focus on enriching our source analysis, rather than assessing changes in microbial composition.

All samples were taken and processed in the same way. The choice of the middle cabin toilet for sampling was based on a study by Christenfeld N in 1995 [9]. Samples were taken with BBL™ CultureSwab™ EZ II (220145, BD, 4000, Belgium), which were moistened with AccuGENE molecular water (BE51200, Lonza, 4800, Belgium) beforehand. Samples were kept at 4 °C and brought to the lab within 1 h.

### 2.2. Total DNA Extraction and Sequencing Library Preparation

Less than 24 h after the sampling, total DNA extraction was performed with the DNeasy Blood & Tissue kit (69506, QIAGEN, 85764, Germany), following the manufacturer’s recommendations. DNA concentration and purity assessment were carried out with a NanoDrop™ 2000 (Thermo Fisher Scientific, Isogen Life Science B.V.,B-4000, Sart-Tilman, Belgium). PCR-amplification of the 16S rDNA V1–V3 hypervariable region and library preparation was performed with the following primers (with Illumina overhand adapters), forward (5′-GAGAGTTTGATYMTGGCTCAG-3′) and reverse (5′-ACCGCGGCTGCTGGCAC-3′). Each PCR product was purified with the Agencourt AMPure XP beads kit (Beckman Coulter; Pasadena, CA, USA) and submitted to a second PCR round for indexing, using the Nextera XT index primers 1 and 2 (Illumina 2018). After purification, PCR products were quantified using the Quant-IT PicoGreen (ThermoFisher Scientific; Waltham, MA, USA) and diluted to 10 ng/µL. A final quantification of each library was performed using the KAPA SYBR^®^ FAST qPCR Kit (KapaBiosystems; Wilmington, MA, USA) before normalization, pooling and sequencing on a MiSeq sequencer using V3 reagents (Illumina; San Diego, CA, USA) [10]. A positive control using DNA from 20 defined bacterial species and negative control were included in the sequencing run. Samples with too low bacterial DNA content or containing PCR inhibitors were not analyzed. Out of the total 96 samples, 81 samples (34 from women and 43 from men) were categorized into 41 Dirty and 40 Clean samples. During the initial processing, we encountered challenges with insufficient DNA yield in 15 of the samples. Consequently, we had to exclude these 15 samples from further analysis, ultimately resulting in a final dataset of 81 samples. This careful curation was essential to ensure the reliability and accuracy of our data analysis, as samples with insufficient DNA could introduce potential biases or limitations in the interpretation of the results.

### 2.3. Bioinformatic Analysis

#### 2.3.1. Microbial Profiling

During the preprocessing stage, the Illumina adapters and primers were removed from the raw data. Subsequently, to ensure data quality and reliability, we applied essential filtering criteria using the command screen.seqs as a trimming process. These stringent filtering steps were crucial for maintaining the integrity of the sequences, as they enforced a maximum allowance of 1 ambiguous base and ensured that all sequences had a minimum length of 450 nucleotides. This curation process provided a solid foundation for subsequent analyses and interpretations in our scientific investigation.

Additionally, following the data curation, we used the MOTHUR software package v1.39.5 (Schloss et al., 2009) to check for chimeric amplification using the VSearch algorithm [11]. The resulting cleaned reads were then aligned to the SILVA database v1.32 [12]. To reduce computational complexity while preserving data representation, we sub-sampled the aligned reads, retaining 10,000 reads clustered into operational taxonomic units (OTUs) using the average neighbor algorithm from MOTHUR v1.39 with a 0.03 distance cut-off [10,13]. This subsampling approach allowed us to efficiently manage the dataset without compromising the accuracy of our taxonomic assignments.

The combined preprocessing and curation steps ensured that our dataset was well-prepared for subsequent analyses, and we are confident in the reliability of our findings. A taxonomic identity was attributed to each (OTUs) by comparison with the SILVA database using an 80% homogeneity cutoff and a threshold of 0.50. The most abundant sequence for each OTU was compared with the SILVA dataset 1.32 version using the BLASTN algorithm to infer species assignment (1% mismatch threshold for specific labeling). Briefly, the species name if known, or the corresponding NCBI accession number was used. Otherwise, for non-identical OTUs, the population was labeled with its corresponding OTU number. In addition to the taxonomic profiling, the OTU representative sequences were used to recover all source information and host-related information from our source-tracking database (based on sequence data).

#### 2.3.2. Source Tracker Analysis

A local 16S rDNA sequence database was built in our laboratory associating sequences to their curated metadata and biotope information.

Data collection stored in our Database

Starting with the 16S rDNA v1.32 set from SILVA database, we removed eukaryotic and vector entries. The corresponding GenBank records of the remaining sequences, containing metadata and study titles, were recovered and curated to keep host and environmental habitat information. In this database, we provide a set of 5 million published 16S rDNA sequences for which taxonomic identity was validated and subsequently labeled with publication and biotope information (host and habitat). Briefly, raw metadata recovered from NCBI entries were reviewed to encode animal and plant hosts with eukaryotic taxonomic affiliation and to keep biotope information. The second layer of global key terms (animal, plant, human, soil, water …) was added. A catalog of annotated metadata terms using a controlled vocabulary was created. This catalog is available at https://github.com/HibaJabri-project/Host_meta_db/blob/master/Host_Dico.obo.zip (accessed on 9 January 2020).

Database design

All datasets were organized using an entity-relationship model [14] using the software package MySQL WorkBench Version 1 (available at https://github.com/HibaJabri-project/Host_meta_db/blob/master/HOST_META_model_database.mwb and accessed on 9 January 2020). All tables are appropriately indexed.

Analysis of restroom data

Sequences are grouped by data partitioning (clustering) according to their similarity and OTUs are defined from a similarity threshold chosen (usually 97%). Using the corresponding accession number, we can deduce the origin source. Bacterial source identification was performed on one side by sequence tracking (ST) (using accession number to find sources) and other side using species name tracking (SNT) as shown in Figure 1. From the final OTU table, populations having known accessions associated with species assignment (keeping the 99% homology threshold) were selected. The corresponding accession IDs were used in the ST approach to recover source tracking keywords associated with these IDs in our source tracking database, returning a reference table containing accession and keywords list pairs. This process was conducted using MySQL query lite version (sqlite3) (query commands available at: https://github.com/HibaJabri-project/Host_meta_db/blob/master/sqlite3_command_restroom.txt and accessed on 9 January 2020).

If the corresponding accessions IDs were not present in our source tracking database, the assigned source labeling was “unknown_source”. Source labeling for OTUs without strict homologous sequences in SILVA 1.32 database (pairwise nucleotide identity below the 99% threshold) was “Not_identical_OTU”.

OTUs or clusters with sequence similarity at the molecular level are used to deduce the source origin. Here, we faced two types of results; one with an accession number corresponding to several sources, and a second accession number corresponding to only one source. In the first case, we valued by giving each source the total account number, fractionating the sum on the total, and then calculating the percentage. However, if the accession number is corresponding to only one source, then we deduce the sum and the percentage of that source. Data are available at this link: https://github.com/HibaJabri-project/Host_meta_db (accessed on 9 January 2020).

The SNT process is based upon a literature search targeting the known biotopes for the list of defined species names obtained from the species assignment protocol during the amplicon profiling analysis. In order to compare both methods (ST vs. SNT), super keywords representing global types of keywords were created for each approach: Animal, Environment, Human, Ubiquitous, Others, and Unknown_source. Ubiquitous was used for the bacterial population whose associated keyword list belongs to several global super keywords. “Animal” was the name used to cover all bacteria associated only with animal biotopes. “Environment” was the name used to cover all bacteria associated with sources like soil, aquatic, and air origin. “Human” was the name used to cover all bacteria associated with human beings. For bacterial populations without any keywords, the super keyword “Unknown _source” was used. Principal Coordinate Analysis (PCoA) was conducted to compare source tracking results for both types of search strategies.

### 2.4. Statistical and Ecological Analysis

For optimal comparison across all samples, the OTU table was rarefied to 10,000 reads per sample used to evaluate ecological indicators (the richness, microbial diversity and Chao1 richness estimator of the samples). Population structure indices like richness estimation (Chao1 richness estimator) [15], microbial biodiversity (Simpson inverse biodiversity index) [16], and population evenness (Simpson evenness index) [17] were calculated using MOTHUR.

The β-diversity was visualized with the Bray–Curtis dissimilarity-based non-metric multidimensional scaling (NMDS) [18] using the *vegan*, *vegan3d*, and *rgl* packages in R [19]. Significant differences between time points were calculated with MOTHUR v1.39 using AMOVA and HOMOVA tests. The AMOVA test is a non-parametric analysis for testing the hypothesis that genetic diversity within each time point is not significantly different from the genetic diversity in all samples together [20]. The HOMOVA nonparametric test analysis was used to test the hypothesis that the genetic diversity within two or more populations is homogeneous [21].

Differences in bacterial relative abundances between gender users were assessed in STAMP tools using a mixed linear model with Benjamini Hotchberg FDR correction for multiple comparisons [22].

## 3. Results

### 3.1. Microbial Profiling

#### 3.1.1. General Characteristics of Microbial Communities

Overall, our study began with 81 DNA samples, which contained a total of 14,915,675 reads. Following the trimming and filtering process, low-quality reads were removed, and all remaining reads had a median length of 495 nucleotides. The reads were then clustered into 17,287 operational taxonomic units (OTUs), with a mean sample coverage of above 97%. The results of the study also revealed a high diversity of microbial biodiversity across the samples, as indicated by a mean index of 15 (Simpson inverse biodiversity index).

According to taxonomic profiling, the bacteria identified in the samples were mainly from four phyla: Actinobacteria (30%), Bacteroidetes (24%), Proteobacteria (22%), and Firmicutes (18%) (Figure 2). The women’s restroom, especially the cabin seat and handle outdoor, was dominated by Firmicutes and Actinobacteria (Figure 2). On the other hand, Bacteroidetes were more abundant in men’s restrooms than in women’s, although the difference was not statistically significant (*p*-value = 0.09579).

The six major genera identified by relative abundance were *Corynebacterium* (23%), *Staphylococcus* (10%), *Cutibacterium* (8%), *Acinetobacter* (8%), *Streptococcus* (4%), and *Lactobacillus* (3%) (Figure 3).

#### 3.1.2. Characterization of Microbial Communities on Different Surfaces of Restrooms

The relative abundance of bacterial communities was compared between two general categories: those found on seat toilet surfaces and those found on surfaces regularly touched with hands (e.g., door handles, cabin in/out, taps, and flush buttons). This difference was driven by several genera whose abundance showed statistical significance (Figure 4b). Regarding the taxa typically associated with surfaces in direct contact with human hand touch (door handles, taps), significant differences were observed in the presence of *Streptococcus* (*p*-value = 6.67 × 10^−3^) and *Cutibacterium* (*p*-value = 7.61 × 10^−4^) compared to urine surfaces (toilet seat) (Figure 4b). On the other hand, *Anaerococcus* bacteria were significantly associated with surfaces in direct contact with water and urine (*p*-value = 2.43 × 10^−4^), particularly toilet seat and urinal flush surfaces (Figure 4b). The cleaning process did not show a direct effect on the presence or absence of bacterial communities but resulted in different microbial population abundances. The microbiota structure of samples before and after cleaning was not globally different (*p*-value > 0.05) according to the AMOVA test, suggesting no segregation of samples according to the cleaning criterion. The Dirty group showed a high presence of *Corynebacterium* on all surfaces, while *Acinetobacter* was more widespread in the Clean group.

Finally, classical indicators of fecal contamination were observed, such as *Streptococcus* and *Enterococcus* spp. on restroom surfaces. Pathogenic microorganisms were present at a low level, including *Staphylococcus* (10%), *Streptococcus* (3.2%), *Enterococcus* (0.6%), and *Campylobacter* (0.2%) (Figure 4a).

#### 3.1.3. Microbial Diversity between Men’s and Women’s Users

In the restrooms, there is a noticeable difference in microbial populations between men and women. Women’s restrooms have a higher abundance of *Lactobacillus*, *Gallicola*, and *Peptoniphilus*, particularly in the cabin seat area. Non-metric dimensional scaling (NMDS) analysis of microbial populations using a Bray–Curtis dissimilarity matrix shows that the samples cluster distinctly by gender. The analysis of molecular variance (AMOVA test) between men and women users confirms the sample clustering with a *p*-value < 0.001. Further, the investigation of Lactobacillus at the species level revealed that *Lactobacillus crispatus* is dominant in the cabin seat area of women’s restrooms. Moreover, OTUs assigned to *Kocuria rhizophila* were found on surfaces related to women’s restroom samples with a *p*-value < 0.005 (Figure 5).

*Staphylococcus* was present on almost all surfaces except for the tap surface of women’s toilets, but this difference was not statistically significant (*p*-value > 0.05) (Figure 6a). Bacterial richness was significantly higher in men’s restrooms than in women’s restrooms (Figure 6b), but there were no significant differences in α-diversity and evenness between the two groups (Figure 6c; *p*-value = 0.87). However, β-diversity differed significantly between the groups (*p*-value = 0.002; Figure 6d).

#### 3.1.4. Bacteria Associated with Animals

To determine the relationship between bacteria and biotopes, a Genbank search was conducted for every known species identified in the microbial profiling using keywords like “Pathology” or “Disease”, “Bacteria”, and “Animal”. Zoonotic bacteria were detected in the bacterial profiling and found on tap-associated surfaces. For example, *Yersinia enterocolitica* species was present on tap surfaces with a very low relative abundance (0.02%) in only one positive sample out of 81 DNA samples. *Haemophilus influenzae* was mostly detected on women’s restroom surfaces, while surfaces such as door handles had a low abundance of zoonotic bacteria like *Erysipelothrix* sp. (0.97%) and *Streptococcus canis* (0.1%). In “Table 2”, several fecal indicator bacteria (FIB) were found to be associated with animal host sources on toilet surfaces based on bibliographic searches. Out of all *Streptococcus* species detected in our samples, only *S. equinus*, a fecal contamination indicator of animal origin, was found on female restroom surfaces.

### 3.2. Source Tracker Analysis

#### 3.2.1. General Characteristics Sources of the Microbial Community in Restrooms

The aim of our investigation was to determine the sources of bacteria in restroom samples. We found that a diverse range of bacterial communities could originate from both human and aquatic sources in restrooms. Using the STN method, we identified the host sources for 76% of the bacterial species found in our samples. The remaining 24% were unknown sources. Our analysis revealed that the bacterial taxa were primarily from environmental sources (54.11%), followed by human (14.48%) and animal (7.24%) sources. Based on the ST method, the six most abundant sources of bacteria in our samples were environment (50.03%), human beings (24.42%), animals (17.36%), other sources (0.78%), ubiquitous sources (0.12%), and 7.29% of unknown sources (Figure 7a).

#### 3.2.2. Bacterial Sources Associated with Animal Hosts

The analysis of animal sources found in restroom surfaces showed that insects were the primary animal sources, followed by bovids, arachnids, mollusks, rodents, suids, and canids, as shown in Figure 7b. Furthermore, the investigation of zoonotic bacteria associated with different animal sources revealed a direct association between Suidae and Equidae keywords and certain bacterial genera, such as *Chryseobacterium, Bergeyella, Fibrobacter*, and *Syntrophococcus*, which were linked to a relatively diverse range of animal hosts, including bovids, insects, equids, and suids.

Interestingly, a comparison between the distribution of sources based on the species level name (nomenclature) and sequence id (accession numbers) showed a considerable difference in the number of source categories using the PCoA analysis (Figure 8). The Principal Coordinate Analysis (PCoA) was utilized to visualize the difference between the number of OTUs obtained using the two types of analyses we conducted, with one providing 556 taxonomic names (Nomenclature sources) and the other providing 3484 accession numbers (Identity sequences sources) in the probable sources in the restrooms. The first component primarily separates human and environmental sequences, while the second component helps identify clusters of animal sources.

## 4. Discussion

In restroom environments, the microbiota is closely linked to the human microbiome, with microbial profiles shaped by various factors such as feeding patterns, hand hygiene, and skin microflora [23,24,25]. Previous studies have explored the concept of defining a bacterial biotope [26], particularly in restrooms [24,27,28]. However, in this study, we employed a new approach and different tools as proof of concept to investigate the direct and indirect contribution of external sources, including animal sources, to the microbial profile of restroom surfaces.

To begin, we conducted a metagenetic analysis to describe the microbial community and identify differences in microbiota between men’s and women’s restrooms, searched for fecal biomarker bacteria, and studied the impact of hygiene cleaning. Our results showed that Actinobacteria were the most abundant phyla in all samples, followed by Bacteroidetes, Proteobacteria, and Firmicutes, which is consistent with other public restroom studies [27,28]. Notably, it is important to mention that samples were collected in March 2017, March 2018, and April 2018 (primarily during the spring, as the first sample was collected in that season). Despite the temporal variations, the relative abundance of these phyla remained consistent across the different sampling periods.

While we found small amounts of Cyanobacteria in our study, previous research reported higher levels, likely due to “Chloroplast” plant material tracked in from outside [28]. However, our study did detect Melainabacteria, a class of Cyanobacteria associated with mammalian gut environments, indicating fecal contamination in some samples [29]. Additionally, we found *Corynebacterium*, a genus ubiquitous in the environment and closely associated with human skin [30], in many of our samples. Using the source-tracking (ST) approach, we identified a variety of sources for *Corynebacterium*, including fish, insects, canids, suids, bovids, farm animals, food staff, and aquatic environments.

Our investigation has revealed the presence of three classes of Cyanobacteria. Melainabacteria, were detected in association with fecal contamination, while Oxyphotobacteria and Sericytochromatia are linked to water environments and were previously identified by Concha et al. [31]. Of the total samples, twenty exhibited a high proportion of *Corynebacterium*, a bacterium widely distributed in various environments [31] and commonly found on human skin [23] based on the “SNT approach.” Flores et al. [27] used Bayesian classifier SourceTracker model tools and Qiime metagenomic tools [7] to demonstrate that the Corynebacterium genus is mostly associated with human skin. Our developed ST method identified multiple sources of the *Corynebacterium* genus group, indicating its ubiquitous nature and potential origins from various sources such as fish gut, insect swabs, canids, suids, bovids, farm animals, food staff, and aquatic environments. The ubiquitous presence of this genus has been well-documented in a prior publication [30], as mentioned in this paragraph.

For the assessment of potential environmental sources of microbiota in the restroom, we decided to focus primarily on animal-origin sources. As a result, we did not include these environmental sources in the final results. However, during the course of our analysis, we did observe the presence of other environmental sources, such as air, soil, and aquatic samples, showing varying proportions across different samples. Although these environmental sources were not the main focus of our investigation, their presence highlights the complexity of the restroom microbiota and suggests their potential contributions to the overall microbial composition. Further research and analysis specifically targeting environmental sources could provide valuable insights into the broader microbial dynamics within the restroom environment.

Overall, our results showed a high diversity of bacterial communities in restrooms, with aquatic and human sources being the primary contributors. We also distinguished between direct and indirect animal sources and found that insect and arachnid sources were present in restrooms as direct contributors, while larger animals were indirect contributors. Our findings are illustrated in Figure 8b. Through our “SNT approach,” we discovered a wide range of bacterial communities in public restrooms, mostly originating from aquatic and human sources such as skin, intestine, and urine, which are commonly found in the restroom environment. However, some keywords overlap, such as bacterial populations associated with fish and the aquatic environment. To address this, we employed the “ST approach” and associated fish source OTUs with aquatic habitats.

The importance of hygiene cleaning in protecting human health from microbial transmission and diseases cannot be overstated. Despite efforts to clean restrooms, studies have shown that they still harbor thousands of types of bacteria and viruses, including common contaminants like fecal bacteria, Influenza, *Streptococcus*, *E. coli*, hepatitis viruses, MRSA, *Salmonella*, *Shigella*, and norovirus [32]. Due to the high number of germs and variables present in restrooms, it may be difficult to remove all contaminants with routine cleaning [33]. Our study found no significant change in microbial profiles before and after cleaning veterinary faculty restrooms (*p*-value > 0.05).

In this study, no significant difference was observed in the microbiota composition before and after cleaning. However, it is crucial to acknowledge that the analysis did not consider the level of contamination, which may have likely decreased over time. It is also important to note that the analysis focused on genetic traces rather than distinguishing between living and dead bacteria. Prior to cleaning, a higher prevalence of Actinobacteria was observed, while Acinetobacter, a microbe typically found in water and aquatic environments [34], increased after the cleaning process. The increase in relative average abundance of Cyanobacteria after cleaning could be due to the removal of other bacteria [32]. The high abundance of Staphylococcus and *Cutibacterium* found in the restrooms, which are typically found on human skin and fecal matter [35,36], may be attributed to the use of tap water and human hands during surface cleaning. It is important to note, however, that not all environmental surfaces in restrooms are cleaned appropriately in some cases [37], and this could also be a contributing factor to the observed bacterial abundance.

The distinct bacterial signatures observed in our findings (Figure 4a) are consistent with those typically associated with healthy urogenital tracts [29,38,39]. The observed differences in bacterial composition between men’s and women’s restrooms may be attributed to sex-related differences in gut microbiota, as previously reported [40,41,42,43,44,45,46,47]. Our findings on bacterial composition between men’s and women’s restrooms are consistent with previous studies [40,41,42] showing that the Bacteroides genus is more present in the male gut. In contrast, we found that *Bifidobacterium*, *Lactobacillus*, *Veillonella*, and *Streptococcus* were more abundant in women. Moreover, our findings align with prior research [45,46,47] in demonstrating that *Lactobacillaceae* bacteria, which are typically found in the healthy vaginal ecosystem, were mostly present in women’s restrooms. Specifically, we observed *Lactobacillaceae* bacteria on seat surfaces in women’s restrooms, which suggests the presence of healthy vaginal microbiota [48,49].

Women’s restrooms also showed the presence of other specific bacterial markers such as *Kocuria rhizophila* (Figure 6a) and *Gallicola* and *Peptoniphilus* (Figure 4b), although in low relative abundance. These bacteria have been associated with urinary tract infections and physiological imbalances in women’s bodies, as reported in previous studies [50].

In our investigation of the Staphylococcus genus, we found that *Staphylococcus* aureus is the leading cause of skin and soft tissue infections [51], despite only accounting for 0.13% of the sequences associated with the genus. Other studies have suggested that *Staphylococcus* spp. may be more prevalent on restroom seats, but our findings indicate that the inside handle door harbors more *Staphylococcus* spp. [52] than the outside handle door, likely due to the hand-washing process and the assumption that hands are dirty upon entering the restroom and clean upon leaving (Figure 4a). Our ST approach also revealed that *Staphylococcus* spp. can be found in not only human users but also in animal or aquatic environments, as well as in gut and skin-human bacteria flora, as seen in other studies [27].

Amplicon sequencing strategies provide a comprehensive view of the microbiota in our samples, without the need for culturing, which can be less efficient in terms of bacterial discovery. In this study, we aimed to identify possible sources of bacteria in the restroom, making culture microbiology unnecessary.

Using the ST approach, we found that the most abundant bacteria sources in veterinary faculty restrooms samples were associated with aquatic (12.32%), human (9.04%), soil (6.87%), gymnosperms (4.84%), fish (4.40%), and food sources (2.60%). Our results differed from those of Flores et al. [27], whose SourceTracker module in Qiime tools [7] relied on a statistical model to estimate source proportions by downloading 10 samples for each suspected source of each group (human, aquatic and soil, and others) and using 16S rDNA bacterial communities. Our local database, based on open public sequence information, provided us with more keyword information about bacterial sources, especially related to organisms, increasing our knowledge from 7.24% to 17.36% of general information associated with animals and reducing the unknown source information compared to the SNT approach based only on literature (Figure 7a). Furthermore, providing information about the source origin was useful for comparing our results with the literature [27,28], in addition to the bacterial profiling analysis.

The interactive coloring of the PCoA plot (Figure 8) reveals a divergence in the resource distribution between the two methods of analysis: one based on taxonomic names (SNT) and the other based on sequence identity (ST). The ST approach, in conjunction with our local database, enabled the identification of a vast array of sources, particularly those linked to animal surfaces in restrooms, which were previously unidentified by the SNT method (Figure 8). Our visualization accurately illustrates the distribution of bacterial sources in restrooms based on sequence identity, highlighting the genuine distribution of bacteria sources in these environments. Notwithstanding, systematic errors associated with metadata and sequencing technologies remain a potential concern. As such, our final results table displays the number of supporting sequences for each cluster enrichment, and our interactive visualization provides the ability to inspect PCoA clusters categorized by source groups (animal, environment, human, ubiquitous, other, and unknown sources). Omitting taxonomy can be advantageous because unknown species of ubiquitous genera are uninformative for source tracking.

Comparing bacterial sources in this study using our local database (ST) with those described in the literature (SNT) facilitated the identification of additional sources associated with ubiquitous species. Furthermore, some zoonotic bacteria species, such as *Yersinia enterocolitica*, *Erysipelothrix*, and *Streptococcus equinus*, were directly linked to swine and horse host animals [53], respectively, based on their presence on restroom surfaces, as previously described in Sullivan 2011 and Sherman 1936 [53,54]. Additionally, known fecal contaminant bacteria found on cabin handles and flush surfaces, such as *Enterococcus cecorum*, were directly linked to the fecal environment, as reported in other studies [55]. Although non-pathogenic bacteria, such as *Erysipelotrichaceae*, are associated with bovid animal hosts through the ST approach and are biomarker bacteria of the animal rumen [56].

The SNT method is efficient for well-known and extensively studied bacterial taxa. However, the ST approach can attribute sources to poorly described or unidentified taxa and can be used in tandem with the SNT method to provide further information on the origin of bacteria. This approach demonstrates a “proof in principle” and validates the significance of sequence-based data in microbial source tracking, with a high probability of yielding correct sources. Nevertheless, some well-known bacteria, such as *Streptococcus canis* [57], were associated with unknown sources due to the absence of source information for sequences deposited in public databases (missing metadata), resulting in the loss of valuable information. To address this limitation, combining the SNT and ST approaches as complementary methods or creating a database with more sequences associated with lesser-known environments and specific biotopes would be advantageous.

Most of the sources in this study were expectedly linked to aquatic, human, and soil environments, as typical of restrooms. However, using a local database and incorporating more information on animal sources provided additional knowledge. We believe that this database could serve as a valuable resource in microbiota profiling, helping the scientific community identify unknown bacteria with regard to their ubiquity or potential biomarker value in key ecosystems. There is a growing interest in the potential use of molecular fingerprinting methods (DNA) not only for detecting but also for identifying contamination sources in various industrial and scientific fields. The insights garnered from this study on bacterial biotopes within veterinary restrooms hold significant potential for detecting the sources of contamination in various research domains, particularly in health sciences and veterinary settings.

## 5. Conclusions

In conclusion, our ST approach proved to be a useful complement to the STN approach, especially when dealing with poorly characterized microbial taxa such as those found in restrooms. By utilizing a local database, we successfully identified discernible differences in the microbiota associated with direct (human microbiome) and indirect (animal) contributions in veterinary restrooms. However, it is important to acknowledge the limitations of our study, including the relatively low sample size and the absence of viability assessment and rigorous sample treatment process (cleaning process). As such, these results should be regarded as an initial exploration, providing a foundation for future research.

To advance our understanding and overcome the limitations, we propose expanding the scope of our study by incorporating comparisons from a wider range of restrooms. In the specific paragraph mentioned, we intended to emphasize that my focus extended beyond the database itself, encompassing factors like sample preparation control, such as assessing the influence of environmental variations before and after cleaning (e.g., temperature or relative humidity). These aspects served as critical starting points for this study.

Furthermore, conducting larger sample sizes with more stringent control measures will yield more robust and comprehensive data, essential for deeper insights into microbial dynamics. These improvements will enable us to establish a solid groundwork for further research in this area. Therefore, the results presented in our current study should be recognized as an initial step, urging future endeavors to address these crucial aspects and enhance the understanding of the subject matter. Notably, improvements to our local database, such as cross-linking with other databases, could help address the issue of missing sequence information.

The insights gained in this study on bacterial biotopes in veterinary restrooms could be beneficial in various research areas, including health sciences and veterinary environments. Comparing the results from veterinary restrooms to those from human hospital environments, public transport stations, or school public toilets could provide a better understanding of the origins of microbial contamination. Source indication labeling could also be used to investigate sources of contamination associated with animal disease, hygiene management in common places, and public health research. By providing a quick and easy way to enrich the metagenetic analysis, this tool has the potential to be widely adopted in many different fields.

## Figures and Tables

**Figure 1 microorganisms-11-02053-f001:**
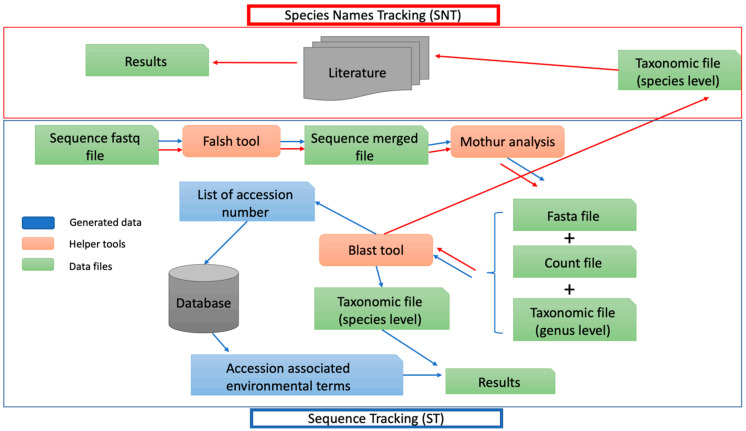
Workflow to analyze restroom data with ST and SNT approach. Microbial profiling analysis was investigated using metagenetic analysis and two kinds of source tracking analysis, one based on the species names (SNT) and the other one based on the frequentist sequence of each OTU to enrich microbial profiling with probable sources of bacterial contamination. Data files in green, helper tools are in orange and data used in the ST pipeline is in blue.

**Figure 2 microorganisms-11-02053-f002:**
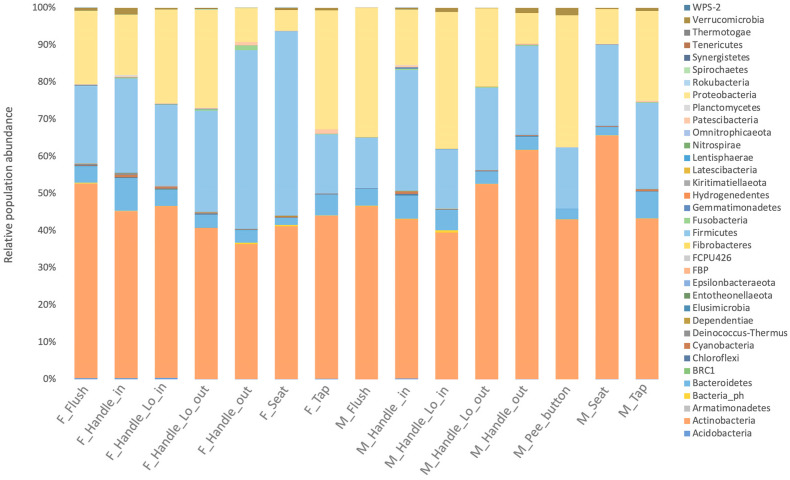
Relative abundances showing the 35 most abundant taxa are shown in phyla for different types of surfaces in men’s user (M) and women’s (F) user restrooms and between surfaces of toilets.

**Figure 3 microorganisms-11-02053-f003:**
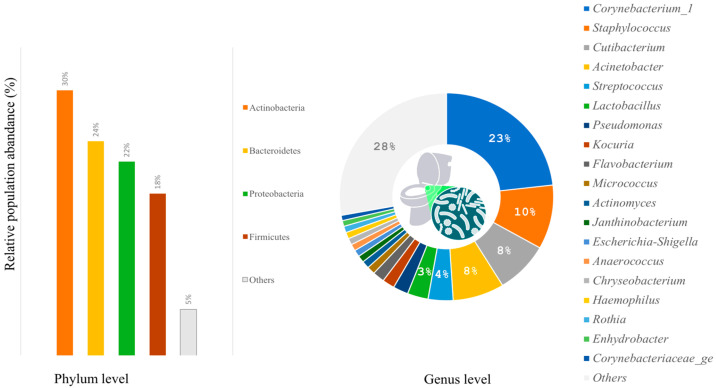
Relative abundances show the 4 most abundant phyla and the 20 most common bacterial genera found on surfaces of toilets.

**Figure 4 microorganisms-11-02053-f004:**
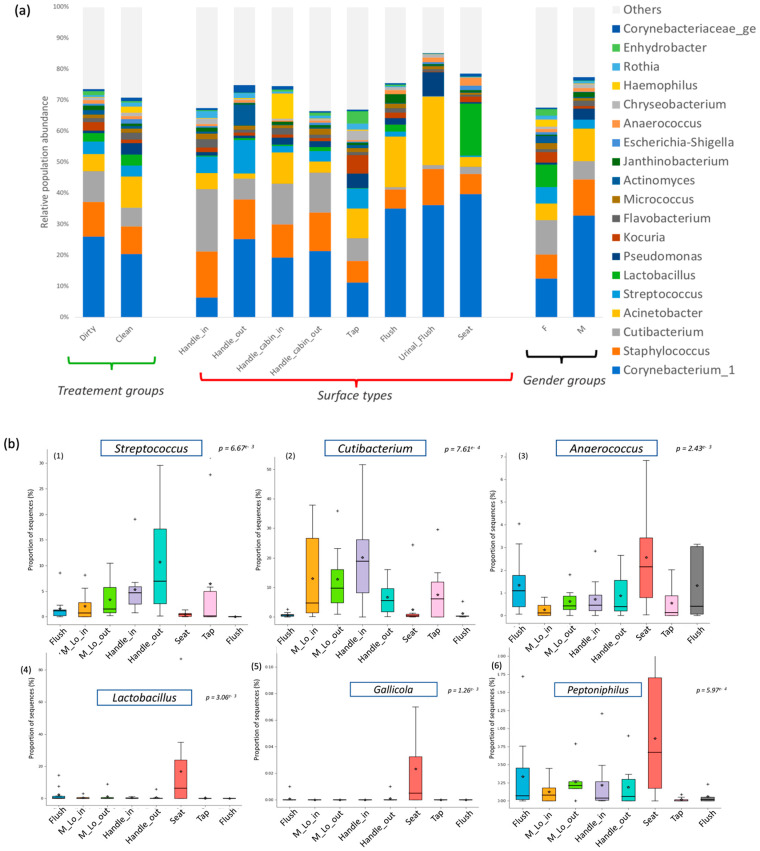
Bacterial diversity with the genus distribution levels expressed as mean cumulative relative abundance in different groups (**a**) Taxonomic composition of bacterial communities (before cleaning process (dirty) or after cleaning process (clean), different sampled locations and different gender users of veterinary faculty restrooms (**b**) Predominance bacterial genera in different restroom surfaces using ANOVA tests. These are default box plot representation of the interquartile range of the relative abundance of the target population in the different groups. Median is indicated as a line inside the box and mean is labelled with a star (*). Outliers values are also indicated (+).

**Figure 5 microorganisms-11-02053-f005:**
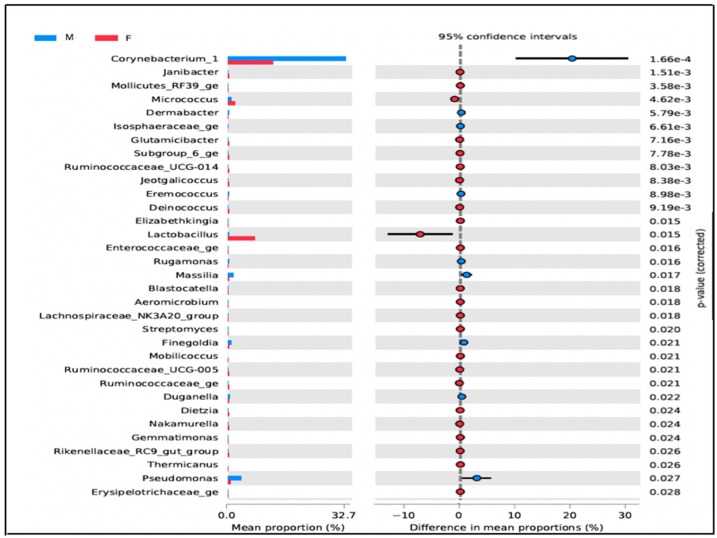
Bacterial Abundance of genera in restroom surfaces for men’s (M) and women’s (F). This figure shows only OTUs with significant differences in abundance between men’s and women’s in veterinary restrooms.

**Figure 6 microorganisms-11-02053-f006:**
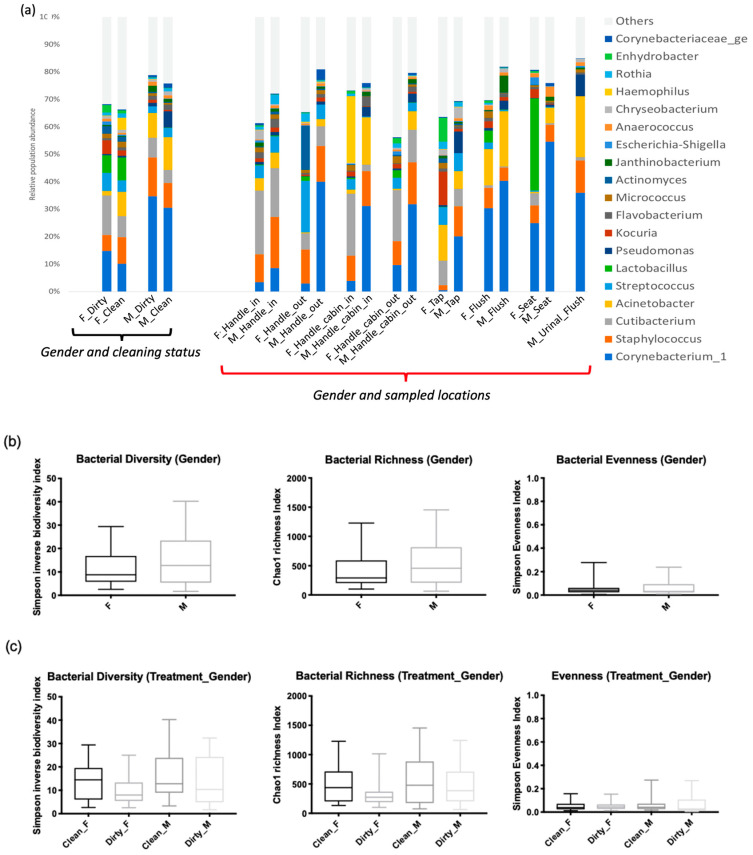
Bacterial diversity between men’s and women’s users. (**a**) Taxonomic composition of bacterial communities using the interaction of two factors (gender user influence before or after cleaning (Gender_treatment)), gender user influence and sampled locations. (**b**) Bacteria diversity (inverse Simpson Biodiversity Index), bacteria richness (Chaol Richness Index) and bacteria evenness (Simpson Evenness Index) both gender users of restrooms (**c**) For both gender users before and after cleaning of restroom surfaces. (**d**) Spatial ordination, of β-diversity between samples deduced by 16S rDNA profiling. Non-metric dimensional scaling (NMDS, k = 3, stress = 0.1) showing standard deviation, men’s user restrooms (M) in red, women’s (F) user ones in black.

**Figure 7 microorganisms-11-02053-f007:**
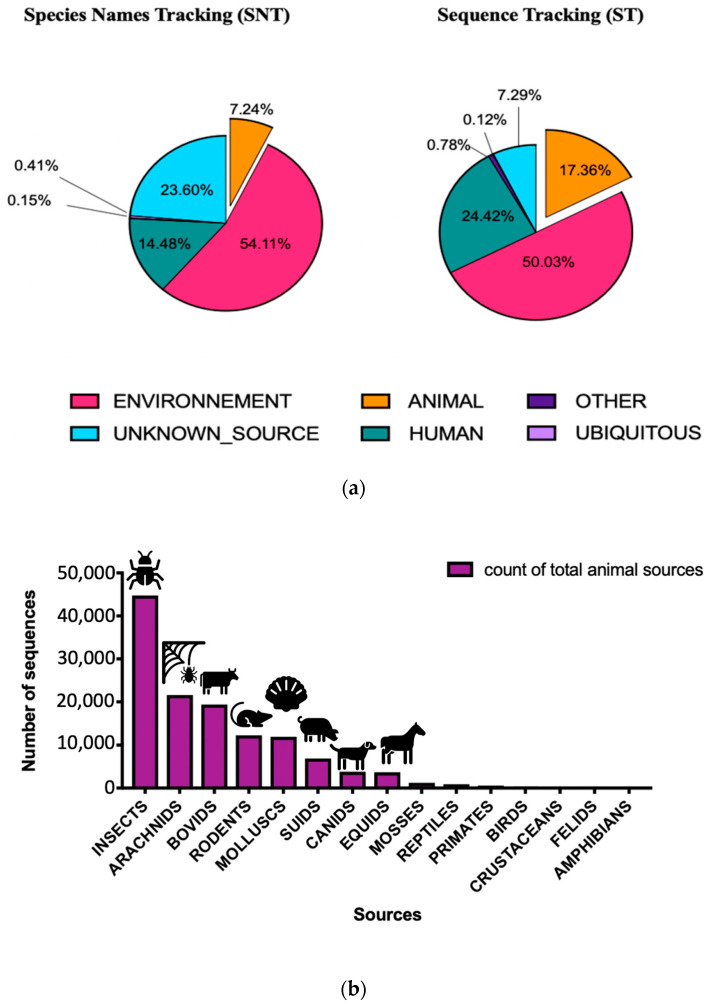
The relative abundance of different taxa in veterinary faculty restrooms with ST and SNT approaches, respectively (**a**) Results of source tracking analysis showing the average distribution of bacterial communities in different surface veterinary restrooms (ST and SNT) (**b**) Details of animal sources with ST approach.

**Figure 8 microorganisms-11-02053-f008:**
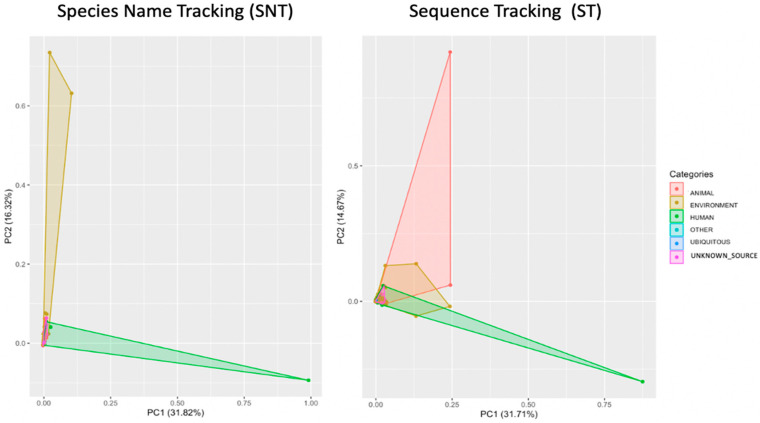
Principle coordinate analysis (PC1 vs. PC2) using PCoA plot (principal components analysis for each species colored by categories ecosystems (animal, environment, human, others, ubiquitous, and unknown sources) with ST and SNT approaches, respectively.

**Table 1 microorganisms-11-02053-t001:** Nature and characteristics of the samples.

Samples	Gender ^1^	Surface Type ^2^	Cleaning Status ^3^
F_Handle_in_Clean	F (Female)	Handle_in	Clean
F_Handle_out_Clean	F (Female)	Handle_out	Clean
F_Handle_Cabin_in_Clean	F (Female)	Handle_Cabin_in	Clean
F_Handle_Cabin_out_Clean	F (Female)	Handle_Cabin_out	Clean
F_Tap_Clean	F (Female)	Tap	Clean
M_Handle_in_Clean	M (Male)	Handle_in	Clean
M_Handle_out_Clean	M (Male)	Hanlde_out	Clean
M_Handle_Cabin_in_Clean	M (Male)	Handle_Cabin_in	Clean
M_Handle_Cabin_out_Clean	M (Male)	Handle_Cabin_out	Clean
M_Tap_Clean	M (Male)	Tap	Clean
F_Flush_Clean	F (Female)	Flush	Clean
M_Flush_Clean	M (Male)	Flush	Clean
M_Flush_M_Clean	M (Male)	Urinal_Flush	Clean
F_Seat_Clean	F (Female)	Seat	Clean
M_Seat_Clean	M (Male)	Seat	Clean
F_Handle_in_Dirty	F (Female)	Handle_in	Dirty
F_Handle_out_Dirty	F (Female)	Handle_out	Dirty
F_Handle_Cabin_in_Dirty	F (Female)	Handle_Cabin_in	Dirty
F_Handle_Cabin_out_Dirty	F (Female)	Handle_Cabin_out	Dirty
F_Tap_Dirty	F (Female)	Tap	Dirty
M_Handle_in_Dirty	M (Male)	Handle_in	Dirty
M_Handle_out_Dirty	M (Male)	Hanlde_out	Dirty
M_Handle_Cabin_in_Dirty	M (Male)	Handle_Cabin_in	Dirty
M_Handle_Cabin_out_Dirty	M (Male)	Handle_Cabin_out	Dirty
M_Tap_Dirty	M (Male)	Tap	Dirty
F_Flush_Dirty	F (Female)	Flush	Dirty
M_Flush_Dirty	M (Male)	Flush	Dirty
M_Flush_M_Dirty	M (Male)	Urinal_Flush	Dirty
F_Seat_Dirty	F (Female)	Seat	Dirty
M_Seat_Dirty	M (Male)	Seat	Dirty

^1^ Gender type; F: toilet female user, M: toilet male user. ^2^ Type surface; all site type used by human hand and skin. ^3^ Cleaning status; Dirty: before cleaning process, Clean: after cleaning process.

**Table 2 microorganisms-11-02053-t002:** Fecal indicator bacteria (FIB) were found in veterinary faculty restrooms and main sources using SNT.

Family	Species	Host-Sources in Bibliography Research (SNT)
*Bacteroidaceae*	*Bacteroides dore*	ANIMAL and HUMAN
	*Bacteroides pyogenes*	ANIMAL
*Bifidobacteriaceae*	*Bifidobacterium merycicum*	ANIMAL
	*Bifidobacterium pseudolongum*	ANIMAL
*Clostridiaceae*	*Clostridium algidicarnis*	ANIMAL
	*Clostridium frigidicarnis*	ANIMAL
	*Clostridium novyi*	ANIMAL
	*Clostridium ruminantium*	ANIMAL
	*Clostridium septicum*	ANIMAL
*Enterococcaceae*	*Enterococcus cecorum*	ANIMAL
*Enterobacteriaceae*	*Escherichia coli*	ANIMAL and HUMAN
*Erysipelotrichaceae*	*Faecalicoccus pleomorphus*	ANIMAL
*Streptococcaceae*	*Streptococcus equinus*	ANIMAL

## Data Availability

All the biosample raw reads have been deposited at the National Center for Biotechnology Information (NCBI) and are available under de Bioproject ID PRJNA810326 under this URL: https://www.ncbi.nlm.nih.gov/bioproject/?term=PRJNA810326 (Registration date: 25 February 2022). Data results are available at this link: https://github.com/HibaJabri-project/Host_meta_db (accessed on 9 January 2020).

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
