# Peer review of "Microbiota Profiling on Veterinary Faculty Restroom Surfaces and Source Tracking"

_microorganisms, 2023, doi:10.3390/microorganisms11082053_

Round 1

Reviewer 1 Report

The manuscript "Microbiota profiling on veterinary faculty restroom surfaces and source tracking" by Hiba JABRI, Simone Krings, Papa Abdoulaye Fall, Denis Baurain, Georges Daube and Bernard Taminiau focuses on testing a local source tracking database to correlate the microbial community, and more specifically microbial taxa, with their sources of origin.

After a careful reading and judgment, I think this manuscript has some major problems that need to be reviewed:

The purpose of the study is unclear. What is its application? Is the purpose of the study just to create a source-bacteria-carrier database?

L. 270-274 and L.422-423 – How did the authors evaluate the quality of the cleanup? For example, the article "The effect of disinfectants on the microbial community on environmental healthcare surfaces using next generation sequencing" (Perry-Dow KA, de Man TJB, Halpin AL, Shams AM, Rose LJ, Noble-Wang JA. Am J Infect Control. 2022 Jan;50(1):54-60. doi: 10.1016/j.ajic.2021.08.027) shows the effect of cleaning products on changes in the microbiome. And different cleaning products affected the microbiome to different degrees.

In addition, in the Discussion section, the authors attribute the lack of change in the microbiome before and after cleaning of veterinary restrooms to poor cleaning quality.  However, the authors should first specify what products were used to clean the rooms.

 L.529 – 530 How do you create a database without sample preparation controls?

 In the Discussion section, authors should be clearer and broader about the application of their research.

Minor comments:

L. 12-13 - it is better to clarify that you are not primarily talking about human bacterial infections of zoonotic origin.

L. 59-60 – «It has already been observed that microbes in restrooms are mostly of human gut origin [7,8].» Frankly, this is a long-known fact that doesn't need to be mentioned.

L. 65-67 – «The advent of high-throughput sequencing techniques allows for the investigation of microbial communities with thorough taxonomical classification using amplicon sequencing strategies.»  - The sentence is not related to the topic of the study

L. 229-231 – «This section may be divided by subheadings. It should provide a concise and precise description of the experimental results, their interpretation, as well as the experimental conclusions that can be drawn.» - This is the text from the journal form that should be deleted

Figure 2b I think this is not necessary information at all. You can see everything in Figure 2A.

Figure 4b - Abbreviated sample names should be made larger

Not all genera and species are italicized

Minor editing of English language required

Author Response

Reviewer #1 (Comments for the Author):

All our responses to the questions and suggestions are highlighted in green, adjacent to each respective one.

The manuscript "Microbiota profiling on veterinary faculty restroom surfaces and source tracking" by Hiba JABRI, Simone Krings, Papa Abdoulaye Fall, Denis Baurain, Georges Daube and Bernard Taminiau focuses on testing a local source tracking database to correlate the microbial community, and more specifically microbial taxa, with their sources of origin.

After a careful reading and judgment, I think this manuscript has some major problems that need to be reviewed:

The purpose of the study is unclear. What is its application? Is the purpose of the study just to create a source-bacteria-carrier database?

Our main objective was to establish a comprehensive database of bacterial sources, and tesded by samples were available at that period of tome. The database was meticulously curated, and extensive testing was conducted using samples collected from indoor restrooms frequently used by veterinary professionals. We believe this approach not only ensures the reliability of our data but also provides valuable insights into the microbial source dynamics within veterinary environments.

Original:

L12- L16: Scientists still wonder if animal-derived microbes are prevalent in the environments of people who handle animals especially when these microbes are suspected of being involved in human disease. The aim of this study was to test a local source tracking database by choosing an environment frequently used by veterinary students and veterinarians and by identifying resident microbiota composition and their sources of contamination (animals, environment, human beings…).

Changes:

L12-L17:In this study, we aimed to develop a comprehensive microbial source amplicon database tailored for source tracking in veterinary settings. We rigorously tested our locally curated source tracking database by selecting a frequently accessed environment by veterinary students and veterinarians. By exploring the composition of resident microbiota and identifying potential sources of contamination, including animals, the environment, and human beings, we aimed to provide valuable insights into the dynamics of microbial transmission within veterinary facilities.

  1. 270-274 and L.422-423 – How did the authors evaluate the quality of the cleanup?For example, the article "The effect of disinfectants on the microbial community on environmental healthcare surfaces using next generation sequencing" (Perry-Dow KA, de Man TJB, Halpin AL, Shams AM, Rose LJ, Noble-Wang JA. Am J Infect Control. 2022 Jan;50(1):54-60. doi: 10.1016/j.ajic.2021.08.027)shows the effect of cleaning products on changes in the microbiome. And different cleaning products affected the microbiome to different degrees.

In addition, in the Discussion section, the authors attribute the lack of change in the microbiome before and after cleaning of veterinary restrooms to poor cleaning quality.  However, the authors should first specify what products were used to clean the rooms.

We added this paragraph in L101: During the cleaning process, a Sani Cud Pur Eco product from the brand Diversey, comprising citric acid and surfactant agents, was used. The cleaner applied the product and added water to effectively clean the restroom.

It is essential to clarify that our emphasis on the cleaning hygiene process was to enable a diverse range of sample testing, rather than specifically evaluating its effect on microbial composition. Our primary objective was centered on exploring potential shifts in microbial sources, with a focus on enriching our source analysis, rather than assessing changes in microbial composition.

270-274 The cleaning process did not show a direct effect on the presence or absence of bacterial communities but resulted in different microbial population abundances.

422-423 Due to the high number of germs and variables present in restrooms, it may be difficult to remove all contaminants with routine cleaning [34].

 L.529 – 530 How do you create a database without sample preparation controls?

. By utilizing a local database, we were able to identify clear differences between microbiota associated with direct (human microbiome) and indirect (animal) contributions in veterinary restrooms. However, given the limitations of our study, including the low sample size and lack of viability assessment and sample preparation control, these results should be viewed as a starting point for future research

My focus was not on the database itself in that paragraph, but rather on the samples tested and their preparation from the restroom (cleaning process), which served as a starting point for this study. To advance our understanding, it would be beneficial to expand the scope and include comparisons from a broader range of restrooms. Additionally, conducting larger sample sizes with increased control measures would provide more robust and comprehensive data for future investigations. This approach will allow us to delve deeper into the intricacies of microbial dynamics and establish a solid foundation for further research in this area.

We have reviewed and approved the paragraph as stated in lines L 569 to L588:

By utilizing a local database, we successfully discerned significant differences in the microbiota associated with direct (human microbiome) and indirect (animal) contributions in veterinary restrooms. However, we must acknowledge the limitations of our study, including the relatively small sample size and the lack of viability assessment and rigorous sample treatment process (cleaning process). These considerations warrant viewing our results as an initial exploration, establishing a foundational basis for future research.

To advance our understanding and address these limitations, we propose expanding the scope of our study by including comparisons from a more diverse range of restrooms. Specifically, the focus of the paragraph in question extended beyond the database itself, encompassing sample preparation control to assess the influence of environmental variations before and after cleaning (e.g., temperature or relative humidity). These aspects served as crucial starting points for our investigation.

 Additionally, conducting larger sample sizes with stringent control measures will yield more robust and comprehensive data, offering deeper insights into microbial dynamics. These enhancements will establish a solid groundwork for further research in this field. Consequently, the results presented in our current study should be acknowledged as an initial step, encouraging future endeavors to address these critical aspects and enrich our understanding of the subject matter. Notably, improvements to our local database, such as cross-linking with other databases, could help address the issue of missing sequence information and further enhance the comprehensiveness of our research.

 In the Discussion section, authors should be clearer and broader about the application of their research.

We appreciate the reviewer's feedback and thank him for his valuable insights. In the Discussion section, we will certainly strive to provide clearer and more comprehensive details about the potential applications of our research.

Our study introduces a novel approach, incorporating different tools as a proof of concept, to investigate both the direct and indirect contributions of external sources, including animal sources, to the microbial profile of restroom surfaces. We firmly believe that the database developed in this study holds significant promise as a valuable resource in microbiota profiling. Its extensive content can aid the scientific community in identifying unknown bacteria and evaluating their ubiquity and potential biomarker value in various ecosystems.

Furthermore, the relevance of our research extends beyond restroom environments, as there is a growing interest in the utilization of molecular fingerprinting methods, particularly DNA-based techniques, not only for detecting but also for identifying contamination sources across diverse industrial and scientific fields. We will elaborate further on these aspects in the revised Discussion section, highlighting the potential implications of our findings and how they contribute to broader applications in the field of microbiology and environmental sciences.

We Changed this line 562: The insights garnered from this study on bacterial biotopes within veterinary restrooms hold significant potential for detecting the sources of contamination in various research domains, particularly in health sciences and veterinary settings.

Minor comments:

  1. 12-13 - it is better to clarify that you are not primarily talking about human bacterial infections of zoonotic origin.

Corrected! And we changed the abstract as mentioned previously.

Original:

L12- L16: Scientists still wonder if animal-derived microbes are prevalent in the environments of people who handle animals especially when these microbes are suspected of being involved in human disease. The aim of this study was to test a local source tracking database by choosing an environment frequently used by veterinary students and veterinarians and by identifying resident microbiota composition and their sources of contamination (animals, environment, human beings…).

Changes:

L12- L17: In this study, we aimed to develop a comprehensive microbial source amplicon database tailored for source tracking in veterinary settings. We rigorously tested our locally curated source tracking database by selecting a frequently accessed environment by veterinary students and veterinarians. By exploring the composition of resident microbiota and identifying potential sources of contamination, including animals, the environment, and human beings, we aimed to provide valuable insights into the dynamics of microbial transmission within veterinary facilities.

  1. 59-60 – «It has already been observed that microbes in restrooms are mostly of human gut origin [7,8].» Frankly, this is a long-known fact that doesn't need to be mentioned.

Has been removed!

  1. 65-67 – «The advent of high-throughput sequencing techniques allows for the investigation of microbial communities with thorough taxonomical classification using amplicon sequencing strategies.»- The sentence is not related to the topic of the study

Has been removed!

  1. 229-231 – «This section may be divided by subheadings. It should provide a concise and precise description of the experimental results, their interpretation, as well as the experimental conclusions that can be drawn.» - This is the text from the journal form that should be deleted

Thank you for your feedback! We have addressed the mentioned issue, and the necessary modifications have been made. The revised version now includes the required changes.

Figure 2b I think this is not necessary information at all. You can see everything in Figure 2A.

Thank you for your feedback! We have addressed the mentioned issue, and the necessary modifications have been made. The revised version now includes the required changes.

Figure 4b - Abbreviated sample names should be made larger

Thank you for your feedback! We have addressed the mentioned issue, and the necessary modifications have been made. The revised version now includes the required changes.

Not all genera and species are italicized

Done!

Reviewer 2 Report

This manuscript investigated bacterial diversity in the veterinary faculty restroom. The different surfaces in male and female restrooms have been collected. Besides, the cleaning hygiene process has been considered as one treat, which is a really interesting topic and can offer valuable insights into the origins of restroom bacteria. It’s valuable work that the authors built a local database about bacteria sources based on GenBank. If they can add an item about zoonoses in that database in the future, it would be better. In this study, the authors got some interesting results, but the design of the experiments was not described well. There are some issues that must be solved before publication.

Here are some comments:

1.     Some sentences in this manuscript are too difficult to understand and may need to be revised. Besides, some sentences missed citations or spaces (such as Line 13, Line 398, and Line 470). Line 189, “silva” needs to be “SILVA”. The scientific name of the bacteria and the name of the genus needs to be italic. The authors may need to double-check the whole manuscript.

2.     Line 93, the authors collected samples in March 2017, March 2018, and April 2018. But this information wasn’t shown in the result and discussion. Moreover, why did the authors decide to collect samples at these time points?

3.     Line 128, how did authors filter 96 samples to 81 samples? There were 30 treatments in Table 1. How did the authors set the experiment? How many replicates were set?

4.     Line 101, in this study, the cleaning hygiene process didn’t affect the diversity of bacteria. It would be better if the authors could add the cleaning methods.

5.     In Table 1, it was M for the toilet male user, but it was H for the toilet male user in the figures.

6.     Line 235, how did the authors trim and filter the data? This may need to be added to the method part.

7.     Figure 2, why were 35 most abundant taxa in Figure 2a but only 4 in Figure 2b? The similar issue in the other figures may need to be revised or add the filtering information.

8.     Figure 3, it was bacterial compassion at the genus level, but there was “Corynebacteriaceae”(family level).

9.     Line 313, “bacteria richness (Chao1 Richness Index)” was not shown in the method part.

10.  Line 325, the authors mentioned zoonoses. What’s the difference in zoonoses abundance between different treatments? This may be added to the discussion part.

Some sentences in this manuscript are too difficult to understand and may need to be revised. Besides, some sentences missed citations or spaces (such as Line 13, Line 398, and Line 470). Line 189, “silva” needs to be “SILVA”. The scientific name of the bacteria and the name of the genus needs to be italic. The authors may need to double-check the whole manuscript.

Author Response

Reviewer #2 (Comments for the Author):

All our responses to the questions and suggestions are highlighted in green, adjacent to each respective one.

This manuscript investigated bacterial diversity in the veterinary faculty restroom. The different surfaces in male and female restrooms have been collected. Besides, the cleaning hygiene process has been considered as one treat, which is a really interesting topic and can offer valuable insights into the origins of restroom bacteria. It’s valuable work that the authors built a local database about bacteria sources based on GenBank. If they can add an item about zoonoses in that database in the future, it would be better. In this study, the authors got some interesting results, but the design of the experiments was not described well. There are some issues that must be solved before publication.

Here are some comments:

  1. Some sentences in this manuscript are too difficult to understand and may need to be revised. Besides, some sentences missed citations or spaces (such as Line 13, Line 398, and Line 470). Line 189, “silva” needs to be “SILVA”. The scientific name of the bacteria and the name of the genus needs to be italic. The authors may need to double-check the whole manuscript.

Thank you for your feedback! We have addressed the mentioned issue, and the necessary modifications have been made. The revised version now includes the required changes.

-We change line 431: The ubiquitous presence of this genus has been well-documented in a prior publication [33]

-Line 13 from the abstract has been changed:

Original:

L12- L16: Scientists still wonder if animal-derived microbes are prevalent in the environments of people who handle animals especially when these microbes are suspected of being involved in human disease. The aim of this study was to test a local source tracking database by choosing an environment frequently used by veterinary students and veterinarians and by identifying resident microbiota composition and their sources of contamination (animals, environment, human beings…).

Changes:

L12-L17:In this study, we aimed to develop a comprehensive microbial source amplicon database tailored for source tracking in veterinary settings. We rigorously tested our locally curated source tracking database by selecting a frequently accessed environment by veterinary students and veterinarians. By exploring the composition of resident microbiota and identifying potential sources of contamination, including animals, the environment, and human beings, we aimed to provide valuable insights into the dynamics of microbial transmission within veterinary facilities.

-We added the reference [9] to line 499

  1. Line 93, the authors collected samples in March 2017, March 2018, and April 2018. But this information wasn’t shown in the result and discussion. Moreover, why did the authors decide to collect samples at these time points?

After the initiation of our database-building process, we decided to conduct additional samples, rendering the year of sampling less critical. It is worth mentioning that samples were collected in March 2017, March 2018, and April 2018 (primarily during the spring, as the first sample was collected in that season). Despite the variations in sample collection over time, we consistently observed a stable relative abundance of Actinobacteria, Bacteroidetes, Proteobacteria, and Firmicutes across the different sampling periods. This temporal stability reinforces the robustness of our findings, indicating a persistent microbial composition within the restroom environment over time. This temporal dimension adds valuable context to our investigation and strengthens the reliability of our database-building approach.

We add this paragraph in line L408: Notably, it is important to mention that samples were collected in March 2017, March 2018, and April 2018 (primarily during the spring, as the first sample was collected in that season). Despite the temporal variations, the relative abundance of these phyla remained consistent across the different sampling periods.

  1. Line 128, how did authors filter 96 samples to 81 samples? There were 30 treatments in Table 1. How did the authors set the experiment? How many replicates were set?

In response to the question about filtering 96 samples to 81 samples, we encountered challenges with insufficient DNA yield in 15 of the initial samples. As a result, we had to exclude these 15 samples from further analysis, leaving us with a total of 81 samples for the final dataset. This decision was made to ensure the reliability and accuracy of our data analysis, as samples with insufficient DNA could introduce potential biases or limitations in the results.

We change paragraph L 135: Out of the total 96 samples, 81 samples (34 from women and 43 from men) were categorized into 41 Dirty and 40 Clean samples. During the initial processing, we encountered challenges with insufficient DNA yield in 15 of the samples. Consequently, we had to exclude these 15 samples from further analysis, ultimately resulting in a final dataset of 81 samples. This careful curation was essential to ensure the reliability and accuracy of our data analysis, as samples with insufficient DNA could introduce potential biases or limitations in the interpretation of the results.

We added this line in L 99: We replicated the samples three times.

  1. Line 101, in this study, the cleaning hygiene process didn’t affect the diversity of bacteria. It would be better if the authors could add the cleaning methods.

During the cleaning process, a Sani Cud Pur Eco product from the brand Diversey, comprising citric acid and surfactant agents, was used. The cleaner applied the product and added water to effectively clean the restroom.

We added this paragraph in Line 101: In the cleaning process, a Sani Cud Pur Eco product from the brand Diversey was utilized, containing citric acid and surfactant agents. The cleaner applied the product and added water to effectively clean the restroom.

  1. In Table 1, it was M for the toilet male user, but it was H for the toilet male user in the figures.

Fixed! Done.

  1. Line 235, how did the authors trim and filter the data? This may need to be added to the method part.

Thank you for your insightful comments. We appreciate your thorough review of our preprocessing steps. As you correctly noted, in the trimming and filter stage, we removed the Illumina adapters and primers from the raw data to ensure the integrity of the sequences. Subsequently, we applied essential filtering criteria using the command screen.seqs(fasta=current, count=current, maxhomop=10, maxambig=1, minlength=450).

The filtering process was designed to rigorously control the data quality, allowing only a maximum of 1 ambiguous base and ensuring that all sequences are at least 450 nucleotides in length. These stringent filtering steps are crucial to minimize potential errors and artifacts, as well as to maintain the reliability of the subsequent analyses and interpretations in our scientific investigation.

Original

Sequence reads were curated using MOTHUR software package v1.39.5 (Schloss et al. 2009) and were checked for the presence of chimeric amplification using the VSearch algorithm [13]. Cleaned reads were aligned to the SILVA database v1.32 [14] and sub-sampled to keep 10,000 reads clustered into operational taxonomic units (OTUs) using the average neighbor algorithm from MOTHUR v1.39 with a 0.03 distance cut-off [12,15]."

Changes in line 147:

During the preprocessing stage, the Illumina adapters and primers were removed from the raw data. Subsequently, to ensure data quality and reliability, we applied essential filtering criteria using the command screen.seqs(fasta=current, count=current, maxhomop=10, maxambig=1, minlength=450). These stringent filtering steps were crucial for maintaining the integrity of the sequences, as they enforced a maximum allowance of 1 ambiguous base and ensured that all sequences had a minimum length of 450 nucleotides. This curation process provided a solid foundation for subsequent analyses and interpretations in our scientific investigation.

Additionally, following the data curation, we used the MOTHUR software package v1.39.5 (Schloss et al., 2009) to check for chimeric amplification using the VSearch algorithm [13]. The resulting cleaned reads were then aligned to the SILVA database v1.32 [14]. To reduce computational complexity while preserving data representation, we sub-sampled the aligned reads, retaining 10,000 reads clustered into operational taxonomic units (OTUs) using the average neighbor algorithm from MOTHUR v1.39 with a 0.03 distance cut-off [12,15]. This subsampling approach allowed us to efficiently manage the dataset without compromising the accuracy of our taxonomic assignments.

The combined preprocessing and curation steps ensured that our dataset was well-prepared for subsequent analyses, and we are confident in the reliability of our findings. We greatly appreciate your input and consideration in improving the clarity and thoroughness of our methods section. If you have any further questions or suggestions, please feel free to let us know. Your valuable feedback is instrumental in enhancing the overall quality and impact of our research.

  1. Figure 2, why were 35 most abundant taxa in Figure 2a but only 4 in Figure 2b? The similar issue in the other figures may need to be revised or add the filtering information.

Thank you for promptly addressing the concern and implementing the necessary corrections to the figure legend. Your swift response and attention to detail are highly appreciated. The improvements made to the figure have significantly enhanced its clarity and accuracy.

  1. Figure 3, it was bacterial compassion at the genus level, but there was “Corynebacteriaceae”(family level).

Thank you for pointing out the issue with the Corynebacteriaceae classification. We encountered challenges in assigning the genus level for this taxon and, as a result, labeled it as "Corynebacteriaceae_ge" instead of "unknown_bacteria" to indicate that it belongs to the Corynebacteriaceae family without further genus-level identification. Despite our efforts to resolve this classification, the lack of sufficient information at the genus level led us to use this interim classification. We acknowledge the importance of accurate taxonomic assignment and will further investigate ways to improve the classification in future iterations of our next study.

  1. Line 313, “bacteria richness (Chao1 Richness Index)” was not shown in the method part.

In line 242 was mentioned that Chao1 Richness Index was calculated using the MOTHER tool.

 We changed this Line L 242: Population structure indices like richness estimation (Chao1 richness estimator) [17], microbial biodiversity (Simpson inverse biodiversity index) [18], and population evenness (Simpson evenness index) [19] were calculated using MOTHUR.

  1. Line 325, the authors mentioned zoonoses. What’s the difference in zoonoses abundance between different treatments? This may be added to the discussion part.

We appreciate the reviewer's comment on the lack of a significant difference in bacteria composition between the treatment processes. Based on our findings, we agree that further exploration of differences in zoonotic bacteria between the treatment groups may not yield substantial insights or add meaningful value to the study. Therefore, we have chosen not to delve into this aspect in our current analysis. We believe that focusing on other relevant and impactful aspects of the research will contribute more effectively to the scientific understanding of the subject matter. Nevertheless, we are grateful for the reviewer's suggestion and have carefully considered its implications in refining the scope of our study.

Comments on the Quality of English Language

Some sentences in this manuscript are too difficult to understand and may need to be revised. Besides, some sentences missed citations or spaces (such as Line 13, Line 398, and Line 470). Line 189, “silva” needs to be “SILVA”. The scientific name of the bacteria and the name of the genus needs to be italic. The authors may need to double-check the whole manuscript.

Reviewer 3 Report

            This manuscript uses a new approach to identify the profile and possible sources of microbiota in veterinary faculty restroom. The manuscript is well-written. My recommendation is to accept after minor revision. My questions and suggestions are shown below.

1.        According to the study results, environmental is an important source of microbiota in restroom. However, the authors did not further discuss the possible environmental source for microbiota in restroom. I suggest that the authors should include a discussion on the possible origins of microbiota from the “environment source” in the discussion section.

2.        The study found no significant difference in the microbiota composition before and after cleaning. My question is, how long after cleaning did the authors collect the samples? Could this sampling method potentially influence the observed difference before and after cleaning? Additionally, were there any variations in environmental factors before and after cleaning, such as temperature or relative humidity (important influencing factors)?

Author Response

Reviewer #3 (Comments for the Author):

All our responses to the questions and suggestions are highlighted in green, adjacent to each respective one.

This manuscript uses a new approach to identify the profile and possible sources of microbiota in veterinary faculty restroom. The manuscript is well-written. My recommendation is to accept after minor revision. My questions and suggestions are shown below.

  1. According to the study results, environmental is an important source of microbiota in restroom. However, the authors did not further discuss the possible environmental source for microbiota in restroom. I suggest that the authors should include a discussion on the possible origins of microbiota from the “environment source” in the discussion section.

Thank you for your valuable feedback. We appreciate your interest in exploring possible environmental sources of microbiota in the restroom. In our study, our main focus was on investigating animal-origin sources, which is why we did not include the potential environmental sources in the final results.

However, during the course of our analysis, we did observe the presence of other environmental sources, such as air, soil, and aquatic samples, with varying proportions across different samples. While these environmental sources were not the primary focus of our investigation, their presence in the data underscores the complexity of the restroom microbiota and suggests their potential contributions to the overall microbial composition.

Given your interest in this aspect, we agree that further research targeting environmental sources could provide valuable insights into the broader microbial dynamics within the restroom environment. We appreciate your thoughtful review and will consider the inclusion of these environmental sources in future investigations to gain a comprehensive understanding of microbial contributions in restroom settings.

We did some changes in line L435 to L444:

For the assessment of potential environmental sources of microbiota in the restroom, we decided to focus primarily on animal-origin sources. As a result, we did not include these environmental sources in the final results. However, during the course of our analysis, we did observe the presence of other environmental sources, such as air, soil, and aquatic samples, showing varying proportions across different samples. Although these environmental sources were not the main focus of our investigation, their presence highlights the complexity of the restroom microbiota and suggests their potential contributions to the overall microbial composition. Further research and analysis specifically targeting environmental sources could provide valuable insights into the broader microbial dynamics within the restroom environment.

  1. The study found no significant difference in the microbiota composition before and after cleaning. My question is, how long after cleaning did the authors collect the samples? Could this sampling method potentially influence the observed difference before and after cleaning? Additionally, were there any variations in environmental factors before and after cleaning, such as temperature or relative humidity (important influencing factors)?

Thank you for your valuable feedback. We appreciate your thorough review of our manuscript and have carefully considered your comments.

We acknowledge your observation regarding the cleaning process in the restroom and the emphasis we placed on it. In line L 101:

In the cleaning process, a Sani Cud Pur Eco product from the brand Diversey was utilized, containing citric acid and surfactant agents. The cleaner applied the product and added water to effectively clean the restroom. It is essential to clarify that our emphasis on the cleaning hygiene process was to enable a diverse range of sample testing, rather than specifically evaluating its effect on microbial composition. Our primary objective was centered on exploring potential shifts in microbial sources, with a focus on enriching our source analysis, rather than assessing changes in microbial composition.

Furthermore, in line L 395, we have provided additional details about the sampling timeline:
Notably, it is important to mention that samples were collected in March 2017, March 2018, and April 2018 (primarily during the spring, as the first sample was collected in that season). Despite the temporal variations, the relative abundance of these phyla remained consistent across the different sampling periods.

We appreciate your attention to detail and would like to clarify that we did not specifically focus on variations in environmental factors before and after the cleaning process. Our primary objective was centered around understanding the microbial sources especially the animal sources within the restroom environment and their potential contributions.

Round 2

Reviewer 1 Report

I am satisfied with the edited manuscript and the authors' responses to the reviewer's comments.

However, I would like to make a few minor comments:

1. L. 124-125 - "...forward (50-GAGAGTTTGATYMTGGCTCAG-30) and reverse (50-ACCGCGGCTGCTGGCAC-30)..." replaced by "...forward (5'-GAGAGTTTGATYMTGGCTCAG-3') and reverse (5'-ACCGCGGCTGCTGGCAC-3')..."

2. L. 334 - "...Bacterial diversitybetween men’s..." replaced by "...Bacterial diversity between men’s..." (There is no space between words).

Minor editing of English language required

Author Response

  1. L. 124-125 - "...forward (50-GAGAGTTTGATYMTGGCTCAG-30) and reverse (50-ACCGCGGCTGCTGGCAC-30)..." replaced by "...forward (5'-GAGAGTTTGATYMTGGCTCAG-3') and reverse (5'-ACCGCGGCTGCTGGCAC-3')..."Thank you for your feedback. I'm pleased to inform you that the requested fix has been successfully implemented in the manuscript.

2. L. 334 - "...Bacterial diversitybetween men’s..." replaced by "...Bacterial diversity between men’s..." (There is no space between words).

Thank you for your feedback. I also fixed it and successfully implemented it in the manuscript.